# Health system capacity and readiness for delivery of integrated non-communicable disease services in primary health care: A qualitative analysis of the Ethiopian experience

**Azeb Gebresilassie Tesema**[1,2]*, **Seye Abimbola**[1,3], **Afework Mulugeta**[2], **Whenayon S. Ajisegiri**[1], **Padmanesan Narasimhan**[4], **Rohina Joshi**[1,3,5‡], **David Peiris**[1‡]

1 The George Institute for Global Health, University of New South Wales (UNSW), Sydney, Australia, 2 School of Public Health, Mekelle University, Mekelle, Ethiopia, 3 School of Public Health, University of Sydney, Sydney, Australia, 4 School of Population Health, University of New South Wales, Sydney, Australia, 5 The George Institute for Global Health, New Delhi, India

‡ Equal senior authors.
* azeb18@gmail.com, atesema@georgeinstitute.org.au

**Data Availability Statement:** All relevant data contributing to the findings are within the paper

## Abstract

### Background

Non-communicable diseases (NCDs) now account for about 71% and 32% of all the deaths globally and in Ethiopia. Primary health care (PHC) is a vital instrument to address the ever-increasing burden of NCDs and is the best strategy for delivering integrated and equitable NCD care. We explored the capacity and readiness of Ethiopia's PHC system to deliver integrated, people-centred NCD services.

### Methods

A qualitative study was conducted in two regions and Federal Ministry of Health, Addis Ababa, Ethiopia. We carried out twenty-two key informant interviews with national and regional policymakers, officials from a partner organisation, woreda/district health office managers and coordinators, and PHC workers. Data were coded and thematically analysed using the World Health Organization (WHO) Operational Framework for PHC.

### Results

Although the rising NCD burden is well recognised in Ethiopia, and the country has NCD-specific strategies and some interventions in place, we identified critical gaps in several levers of the WHO Operational Framework. Many compared the under-investment in NCDs contrasted with Ethiopia's successful PHC models established for maternal and child health and communicable disease programs. Insufficient political commitment and leadership required to integrate NCD services at the PHC level and weaknesses in governance structures, inter-sectoral coordination, and funding for NCDs were identified as significant

and in supplementary file. Others potentially identifiable data are stored on a secure network and cannot be shared publicly in order to protect participant confidentiality and to adhere to the permission obtained from the University of New South Wales (UNSW) Human Research Ethics Committee.

**Funding:** The project was supported by the George Institute for Global Health, Australia through the Seed Grant funds dedicated for under-served populations in LMICs for 2019/2020. The UNSW Scientia Scholarship program supports AGT and WA. SA was supported by the Australian National Health and Medical Research Council (NHMRC) through an Overseas Early Career Fellowship (APP1139631). RJ is supported by the Australian National Heart Foundation (APP 102059) and a UNSW Scientia Fellowship. DP is support by NHMRC career Development Fellowship, Level 2 and Australia National Heart Foundation Future Leader Fellow. The funders had no role in study design, data collection and analysis, decision to publish, or preparation of the manuscript.

**Competing interests:** The authors have declared that no competing interests exist.

barriers to strengthening PHC capacity to address NCDs. Among the operational-focussed levers, fragmented information management systems and inadequate equipment and medicines were identified as critical bottlenecks. The PHC workforce was also considered insufficiently skilled and supported to provide NCD services in PHC facilities.

## Conclusion

Strengthening NCD prevention and control through PHC in Ethiopia requires greater political commitment and investment at all health system levels. Prior success strategies with other PHC programs could be adapted and applied to NCD policies and practice, giving due consideration for the unique nature of the NCD program.

## Introduction

In 2019, non-communicable diseases (NCDs) accounted for about 74% and 32% of total deaths globally and in Ethiopia [1]. Primary health care (PHC) is a vital instrument to achieve universal health coverage (UHC) and address the ever-increasing NCD burden [2]. PHC provides a decentralised healthcare platform and is the best strategy for delivering integrated and equitable NCD care [3].

Ethiopia's PHC system, which has been in place for the past three decades and hailed as a model in the African region, has enabled the country to improve health service coverage and attain better health outcomes in maternal and child health (MCH) and communicable diseases [4–6]. The achievements were built on a decentralised PHC model foundation, implemented primarily through a community-based health extension program (HEP) [7, 8]. Box 1 provides a summary of the structure of the health system.

### Box 1. The Ethiopian health system in context

Over the past three decades, PHC has been the guiding principle of the Ethiopian health system [4, 5]. The country has a three-tiered health care system. It consists of primary, secondary, and tertiary levels of health care and a decentralised structure that has devolved core responsibilities and decision making to sub-national units. The primary level of care includes primary hospitals, health centres and health posts (the lowest-level health system facility, at village level). The primary health care unit comprises five satellite health posts and a referral health centre. [8–11] (Fig 1).

Ethiopia implemented the Health Sector Transformation Plan I (HSTP-I) as a national health sector strategy from 2015/16-2019/20, and the second phase, HSTP-II, will be in effect from the year 2020–2025. NCDs were considered one of the major disease control priorities in the HSTP-I and a national strategic action plan (NSAP) for the prevention and control of NCDs was developed in 2014 [8, 12]. Furthermore, the country has also developed other disease-specific strategies and initiative, such as the National Mental Health Strategy, the National Cancer Control Plan, and the National Framework Convention on Tobacco Control [13].

While the NSAP is currently under revision, it sets out four major priority areas: (1) strengthen national response through policy, governance and leadership; (2) health

promotion and disease prevention that halt the risky behavioral factors; (3) quality and comprehensive treatment and care through clinical and non-clinical teams; and (4) improving monitoring, evaluation and use of evidence [12]. One of the operational focus areas of the NSAP has been to strengthen and deliver NCD preventive and curative services through the PHC system [12]. The country's Health Extension Program (HEP) has been recently revisited so that health extension workers (HEWs) can undertake NCD prevention and promotion activities. The HEP is a community-based strategy implemented since 2003 to deliver basic health promotion, disease prevention and selected curative health services at the community and health post level [7]. Recently, the government also introduced a health facility governing board to facilitate linkages between the health system and the community and improve community accountability in service provision. The PHC unit board has decision-making powers; it approves facility plans, budgets, and monitors programs' implementation [14].

However, despite a national NCD strategy and some high-level interventions, progress in providing an integrated NCD service has been very minimal, particularly at the PHC level [12, 13, 15]. Access to and provision of NCD services varies widely between facilities, urban and rural settings, and disease types [15, 16]. Only 46% and 34% of all health facilities in the country are ready to offer diabetes and cardiovascular diseases management services in 2018 [15].

Thus far, there have been few empirical studies to appraise the health system's capacity and readiness for NCD services [17]. This is despite the fact the national strategy on NCD has been in place for over five years now. The few studies that exist are focused on NCD epidemiology, including its risk factors, with little attention on barriers and enablers of service availability within the PHC system [15–17].

Given the rising NCD burden, this study aims to determine the capacity and readiness of Ethiopia's PHC system to deliver integrated and people-centred NCD services. Specific questions we seek to address include: (1) what are the main strategic and operational levers needed to facilitate the implementation of NCDs services in PHC? (2) how can the health system build on the past success stories and lessons to integrate NCD prevention and care? And (3) what additional enabling elements already exist in the system that can be harnessed and capitalised on for optimal NCD service integration into PHC service delivery models?

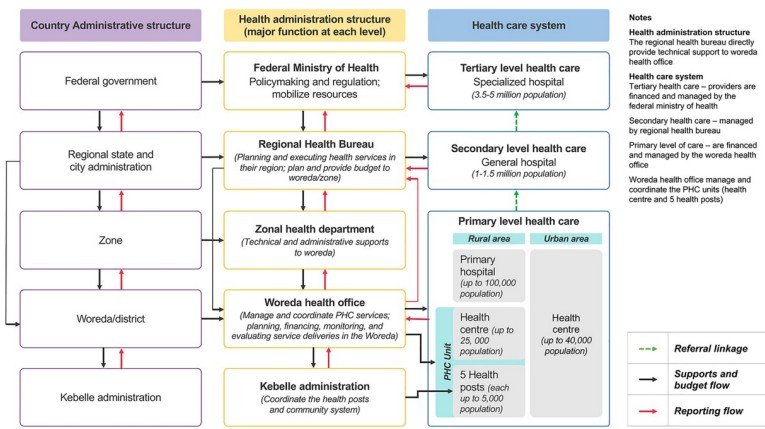

**Fig 1. Ethiopian health care system.** Source [9, 11, 20].

## Methods

### Study design

We conducted a qualitative study from August to October 2019 to explore the capacity and readiness of the PHC system and identify levers to accelerate progress for the delivery of integrated NCD services in the PHC system in Ethiopia.

### Study settings

The study was conducted in Ethiopia. National-level policy makers were selected from the Federal Ministry of Health (FMoH), in the capital city of Addis Ababa. Two regions, Tigray Regional state and South Nations, Nationalities and Peoples Region (SNNPR) were selected purposively for interviews with regional health leaders. We selected five woredas, equivalent to district (hereafter stated as woreda), and two cities from the two regions to facilitate interviews with woreda health office managers and PHC workers. The areas chosen from Tigray regional state were, AtsebiWemberta, Hawzien and KilteAwla'elo and Mekelle Special zone. Similarly, Kachabira and Hadaro Tunto Zura woredas and Hawassa city were included from SNNPR. The areas were selected in consultation with the respective regional health bureau officials and based on their PHC (mainly HEP) performance and accessibility during fieldwork.

### Participant recruitment and data collection

The qualitative data were obtained through semi-structured interview guides developed by reviewing different national strategic documents and were further refined during data collection (S1 File). Key informant interviews were held with national and regional policymakers, and in-depth interviews were carried out with managers and PHC workers.

We interviewed twenty-two participants, purposively recruited from different background. These included nine national and regional level policymakers, a participant from partner organization in Ethiopia, four woreda health office managers and coordinators and eight PHC workers, including HEWs. The PHC workers were recruited from health facilities that were part of a related quantitative study in the two regions. Recruitment emails were sent out to policy maker participants to involve in the study and they were included based on their role and relevant positions during the time of data collection. While, PHC workers who were directly involved in service delivery were involved. See Table 1 for participants' details.

The primary investigator (PI- AGT) conducted the key informant and in-depth interviews with local research assistants' assisting in the recording, note-taking, and prescribing of the interviews. All interviews were carried out face-to-face and lasted an average of one hour. The interviews were held in the place and time of each interviewee's choice to avoid interference with the regular duty of participants and ensure the interviews were held in a private space. After each interview, the PI, a research member (AM) and research assistants discussed emerging issues, took notes and revised the tools for subsequent interviews. This process also ensured data saturation. Interviews were conducted in Amharic and Tigrigna and transcribed to English verbatim by local research assistants and the PI.

### Conceptual framework and data analysis

The PI re-reviewed all the transcripts to ensure that the material has been appropriately translated, keeping the original conversations recorded in the interviews. Transcription and translation accuracy was checked by transcribing and translating few interviews by the PI and comparing the contents with those completed by the research assistants. Data familiarisation

**Table 1. Qualitative study participant characteristics, Ethiopia, 2021.**

| Participant(P) code | Region/location | Study participant's working area | Age category |
|---|---|---|---|
| Participant 1 (P1) | Addis Ababa | Federal ministry of health (FMoH) | 31–35 |
| P2 | Addis Ababa | FMoH | 46–50 |
| P3 | Federal | FMoH | 31–35 |
| P4 | Tigray Regional State | Regional health bureau (RHB), Tigray region | 36–40 |
| P5 | Tigray Regional State | Woreda health office head, Tigray region | 36–40 |
| P6 | Tigray Regional State | Woreda health office head, Tigray region | 46–50 |
| P7 | Tigray Regional State | Primary hospital chief executive officer, Tigray region | 31–35 |
| P8 | Tigray Regional State | Health centre nurse in Tigray region | 46–50 |
| P 9 | Tigray Regional State | Health extension worker (HEW) from Tigray region | 31–35 |
| P10 | Tigray Regional State | Health centre director in Tigray region | 41–45 |
| P11 | SNNPR | Regional health bureau, SNNPR | 41–45 |
| P12 | SNNPR | Regional health bureau, SNNPR | 41–45 |
| P13 | Federal | FMoH | 36–40 |
| P14 | SNNPR | Woreda health office head, SNNPR | 26–30 |
| P15 | SNNPR | HEW from SNNPR | 31–35 |
| P16 | SNNPR | Health centre director, SNNPR | 26–30 |
| P17 | Tigray Regional State | HEW from Tigray region | 36–40 |
| P18 | Tigray Regional State | Regional health bureau, Tigray region | 51–55 |
| P19 | Addis Ababa | Partner organization, Ethiopia | 46–50 |
| P20 | SNNPR | Primary hospital, Medical director, SNNPR | 26–30 |
| P21 | Addis Ababa | FMoH | 31–35 |
| P22 | SNNPR | Woreda health office head, SNNPR | 26–30 |

was done prior to the start of data coding. The PI is a native speaker of both languages used in the interviews.

The transcripts were coded and organised using NVivo 12 software (QSR International, Vic). Three interviews were coded independently by a research member (RJ) for cross-checking. The researchers' codes were then discussed and compared among research members to check the meaning and patterns expressed by study participants. Both inductive and deductive methods were employed in data analysis. Members made consecutive discussions to identify emerging areas of importance through a thematic approach. We also used as many codes during the initial coding process and later combined those with similar themes into categories. Candidate themes that emerged from analyses were first indexed using the World Health Organization's (WHO) health system building blocks [18].

The findings were subsequently framed into thematic and sub-thematic areas, using the newly launched WHO Operational Framework for PHC systems [19].

The WHO PHC framework proposes 14 levers to translate the Astana Declaration's global commitments, which was endorsed in 2018, into actions and interventions to accelerate progress towards strengthening PHC oriented systems [19]. We aligned our key findings with the strategic and operational levers in this framework. Strategic levers represent higher level factors that are considered pre-requisites for the operational levers and cover four sub-themes: political commitment and leadership at national and sub-national levels; governance and policy framework; funding and allocation of resources and community engagement and other stakeholders. The operational levers in the WHO framework covers ten sub-themes, however, for conciseness we condensed these into the following four: care delivery models that enable integrated NCD services; PHC workforce; physical infrastructure in PHC facilities, medicine and equipment supply; and monitoring and evaluation.

### Ethical considerations

We obtained ethics approval from The University of New South Wales (UNSW) Human Research Ethics Committee (HC190014) Sydney, Australia and National Research and Research Ethics office under the Ethiopian Public Health Institute (EPHI-IRB-194-2019), Ethiopia. The Federal Ministry of Health and the two regional health bureaus provided required support letters for the research. Each woreda office also provided a support letter to conduct the study in the selected health facilities. Respondents were informed about the aim of the research, and all participants gave their written consent to involve in the study. No participant has dropped out or refused to participate. To ensure confidentiality and anonymity of the data, we replaced participants' names with codes during data analysis and presentation (Table 1).

## Results

### Theme 1: Core strategic levers for primary health care delivery

**1. Political commitment and leadership at national and sub-national levels.** Key government actors, at various levels, in Ethiopia demonstrate a high level of political commitment and leadership in putting PHC at the centre of the country's health system agenda. A national NCD strategy has been in place since 2014 as a response to the rising burden of NCDs, and this was accompanied by several high-level initiatives to address NCD risk factors. However, participants mentioned leadership gaps in contextualizing, translating, financing and implementing existing strategies into PHC by federal and sub-national governments (regional health bureaus and woreda managers). The leadership gap, in turn, led to limited attention for NCDs at all levels and was cited as a significant factor for the poor implementation of NCD programs at the PHC level. Also, as per participants' opinions, gaps were seen in prioritizing NCD programs during planning, resource allocation, supportive supervision, and monitoring and evaluation.

*"The minimal attention given for NCDs starts from the top level. It would have been good if NCDs were given equal emphasis like (for example) the malaria program because the incidence of diabetes is actually higher than malaria."* Woreda health officer from Tigray region (P5).

The health centre director from SNNPR (P16) concurred: '*We focus on maternal health issues. It is only recently [that] the Government is giving attention [for NCDs]*'. The perspective from lower-level officials contrasted with higher-level officials who perceived that action was being taken, '*At [the] higher level there is no issue. Comparing to previous years, the Government is giving some attention*' (P1). The gaps were explained in terms of decision-making space and limited technical capacity at the lower levels to adapt national plans and design implementation strategies appropriate to local conditions. Again, there was a disconnect between lower and higher-level perspectives on this issue. Local actors expected better support from higher-level officials; however federal officials perceived that sufficient support was being given and that lower-level officials needed to be more proactive.

*"Those who are at the bottom wait for changes to come from high-level. There is a problem in (them) being innovative and doing what is required on the ground. . .. So, we in collaboration with higher leadership are cascading the program and direction to the lower levels."* Federal MOH official (P13).

**2. Governance and policy framework that ensure NCD service delivery at the primary health care.** Although there is a dedicated NCD prevention and control unit with its staff at the federal level, participants commented on a high degree of structural variation at regional and other lower health system levels. A participant from the FMoH (P13) said;

*"If you go down from the ministry to regional, zonal and woreda levels of the health system, it is highly fragmented. . . in some areas, the structure is non-existent, whereas in others there is an organised and functional structure."*

In some regions, the NCD program was coordinated through a separate department, in some others at a case team level and in others with an NCD focal person only. A participant from Tigray RHB (P4) said: *'we have an NCD focal person at the RHB, but we are yet to assign an NCD focal person in the woredas and at health facilities.'* However, according to participants from FMoH (P13, P1), there were regions without any structure for NCD programs, and other program coordinators jointly managed NCDs.

The weaknesses of sub-national NCD structures were linked in part to capacity constraints. A participant from FMoH (P1) explained: *'. . . there is a good start at higher leadership levels. . .as NCDs are included in the national policy/strategies; however, the level of preparation at the lower level is deficient.'* As highlighted by one regional health bureau participant, there also appears to be a perceived lack of mandate to integrate NCDs into service delivery *'. . .the woreda structure is not capable of carrying the load of the low-level health activities, which means the people working there were not capable enough to plan and execute effectively . . .'* (P4).

**3. Availability of funding and allocation of resource for NCD program.** Participants noted several funding challenges including: a lack of financial protection leading to high catastrophic household out-of-pocket (OOP) expenditure related to NCD care; limited government funding for PHC facilities with a disproportionate allocation of scarce resources for NCDs to tertiary care facilities; and procurement of large expensive technologies again directed toward the tertiary care sector. Even in the PHC sector, as a participant from the SNNPR woreda health office (P14) explained, *'much of the funding goes to maternal and child health, as the services are free for this group.'*

A participant from a partner organization, Ethiopia (P19), added that, *'government support for NCDs is almost nil; it focuses more on HIV, tuberculosis, malaria and the like.'* However, a higher-level policy maker (P1) noted that this does not necessarily limit its implementation as *'funding from HIV, tuberculosis and malaria programs, can be optimized to support other services.'* However, the challenge is that *'if there is a shortage of fund in the system, NCD programs would be the first to be affected.'* (higher-level policymaker; P1).

**4. Community and other stakeholders' engagement.** Multisectoral engagement and coordination, capable of addressing cross-cutting issues related to NCD programs were considered weak. A participant from a partner organization (P19) said that *'The NCD program is hanging at the ministry of health level with limited interdepartmental and intersectoral coordination.' 'Everyone runs in their directions, (with) no significant and sustainable results for the community.',* added a participant from the Federal Ministry of Health (P3).

Regarding community engagement, each health facility has a governing board representing community interest and hold the health system accountable to the work done (or not done) in their locality. However, as participants mentioned, historically, the board focuses on MCH program and facility budget utilization and doesn't prioritise and promote NCDs. A regional hospital CEO said (P7), *'The coordination within the community would have been stronger if the governing board had paid sufficient attention to NCDs because any task which gains their attention always gets implemented.'*

## Theme 2: Operational levers for primary healthcare that enable NCD service delivery

**1. Model of care that enables integrated NCD services at PHC.** Participants considered that NCD service delivery is generally disconnected from community care and implemented as a vertical program concentrated in hospital facilities. *'Most NCD activities are vertical programs. We don't have strong horizontal NCD programs.', noted* a participant from Tigray RHB (P4). Besides, participants noted that most facilities are not ready to provide NCD services: '*I do not think our PHC units are ready to provide NCD services as they lack required inputs, including human resources.'(P4).* Health facility participants mentioned that services at health centres are limited to basic screening and diagnosis services for hypertension and diabetes. *'HEWs only provide NCD preventive education in the health post and during house-to-house visits, but this is not structured and not on as regular a basis as we do for other services.'* a HEW (P15) explained.

The referral system in place was seen as a major limitation of the current NCD service delivery model. Patients often tend to self-refer to specialists and higher-level facilities due to an absence of NCD services in the health centres and primary hospitals. Even in some cases, where services are available, the community has lower acceptance for the services provided in these facilities. A lack of referral feedback from the secondary and tertiary levels of care created further gaps in the follow-up of care for patients at the PHC facilities. *"Patients are supposed to get services at the health centre but are bypassing it to go to the hospitals thereby creating more burden to the higher-level health care system. . .'* a participant from FMoH (P13). *'. . .the problem is the PHC facilities are not receiving feedback and follow up is weak.'* CEO of a primary hospital from Tigray (P7).

Several participants suggested lessons learnt from HIV, tuberculosis and malaria models of care can serve as exemplars for good practice of integrated service delivery. These models could be adopted as a strategy for effective NCD service delivery. *'The HIV Antiretroviral Therapy (ART) service has been successfully implemented at the health centre level through the decentralization of health services. As such, a lesson can be applied for NCDs'* said RHB participant from SNNPR (P11).

**2. Primary health care workforce for NCD service delivery.** Participants described gaps in having support mechanism and a clear strategy for NCD-related training at the PHC level. '*There is no specific training for NCDs. . . [health workers] didn't get update[d] information other than the information that they get from internet and textbooks by their effort.', a* woreda health office head lamented (P14). This mainly created knowledge and capacity gaps among PHC workers, including the HEWs—as discussed by the PHC workers participants. According to an official from the FMoH (P2): *'the capacity of the HEWs to provide health services is not sufficient. There is a gap in knowledge and preparation to teach the community, especially for the NCD program'.* Participants perceived that such knowledge and skill gaps are even seen among HEWs who have already undergone Level IV training. However, the higher-level participants and health facility directors noted current initiatives being in place by the ministry of health, including the initiatives to post higher-level medical staff and provide in-service training for nurses and other health-care workers:

*"We have only health officers and nurses in our health facilities, so yes there is a knowledge gap. But now, with the support from top management, we are deploying medical doctors to the health centres and anticipate the knowledge and skill gaps will be minimized."* Participant from FMoH(P2).

One woreda bureau head from Tigray region (P5) explained: '*the woreda management including the woreda head or deputy head and representatives from each case team provide*

*regular supportive supervision (expert wise support) every three months to PHC units'*. However, a facility level participant (P9) described challenges related to these supervisors' limited capacity and low priority accorded to NCDs as follows: *'. . . the woreda staff don't get any refresher training on NCDs and similarly, I didn't receive any training on NCDs'*.

**3. Medicines, equipment and infrastructure.**   Most participants described the shortage of medical supplies, bureaucracy around procurement and distribution of medicines, and inadequate funding among the main bottleneck for delivering NCD services in different settings. While there are ongoing initiatives to overhaul the procurement and provision of NCD equipment nationally, critical resources are still concentrated at large hospitals. A regional health bureau participant (P4) said: *'We have a limited number of PHC facilities with functional Blood Pressure apparatus, glucometers and other equipment, which impacts the implementation of NCD care'*. A nurse working in a health centre (P8) added: *'The [consultation room itself is not adequate. I mean, they (patient with NCDs) need privacy and a quiet place.'* One primary hospital CEO (P7) participant recognised this as a problem and mentioned some health facilities had redesigned existing rooms: *'We opened a separate chronic follow up clinic (for diabetes and hypertension).'*

**4. Health information systems for NCD service delivery.**   One participant explained, *'The health management information system (HMIS) for NCDs is fragmented across all levels and facilities, and we recognize this as a problem'*(P13). As most participants pointed out, in most PHC units, NCD data are not properly recorded and used for decision making due to a lack of uniform systems to capture NCD data. A participant from SNNPR regional bureau (P12) emphasised that *". . .NCD cases are missed during registration and reporting. There are also problems linked to compiling, analysing and use the data to monitor and evaluate performance. . .'*

Also, as another policymaker (P1) explained, *'until recently, NCD indicators were not part of the core HMIS list'*, *and* is seen as a challenge. A participant from the FMoH (P3) indicated: *'unless we improve the information system for NCDs, we cannot know the burden of disease and it is difficult to improve service coverage and quality.'* Furthermore, as health facility participants described, the lack of NCD performance indicators contributes to the NCD program's limited attention within PHC. *'The key performance indicator (KPI) is one of the guides that helps us to focus and give direction to our works, but there is no single NCD indicator on it.'* Health centre director, SNNPR (P16). A participant from the hospital (P7) also added, *'we didn't evaluate NCDs during KPI assessment, and they are not part of the monitoring framework.'*

Increased use of digital technologies for the day to day activities of HEWs was highlighted by one HEW participant (P17) from Tigray: *''During my house-to-house visit, if I have a mobile loaded with health application, the household member tend to gain relatively more interest and better knowledge about the topics.* However, *'Few HEWs have mobile tablets supported by the partner organizations.'* Woreda participant from SNNPR (P14) noted. Another HEW from Tigray (P17) explained: *'if such a system was introduced for NCDs, this could disseminate information very clearly and in a short time to the family and I can tell you I am interested to do it too'*.

## Discussion

Ethiopia's PHC system is widely praised and often seen as a model for low-income countries [5]. However, in this study, we found that political commitment to act and leadership in integrating NCD services at the PHC level were weak. Despite a national strategy on NCDs and existence of some interventions on NCD risk factors, interview participants identified considerable gaps in governance structure, inter-sectoral coordination, and funding for NCDs. NCD service readiness at PHC units was low: the workforce was minimally trained on NCDs; there were limited equipment and medicines, and the program monitoring framework did not

integrate NCD indicators effectively. We discuss the implications of these findings below, drawing on relevant literature and the WHO PHC Operational Framework [19].

Ethiopia's prior success in PHC reforms, primarily in the areas of MCH, HIV and malaria, were driven by strong political leadership, good governance, resource commitments, effective stakeholder engagement, and the existence of a well-defined program of action [4, 5, 20]. The establishment of national and sub-national multi-sectoral committees and the involvement of sub-national political leaders and community members played a crucial role in the success of HIV and MCH programs in Ethiopia [5, 13]. Such high-level, whole-of-government, multi-sectoral bodies, chaired by high-level officials, were active in policy coordination, program design and resource allocation [13].

However, participants in our study noted that at the PHC level, NCD programs did not receive the same commitment, structural support, stakeholder engagement and funding as MCH, HIV and other communicable diseases programs. The lack of enthusiasm for NCDs is in part linked to a mindset informed over many years by health system managers and implementers that programs such as HIV and MCH are a higher priority than NCDs. This is further exacerbated by the limited awareness of the changing burden of diseases among sub-national actors and the general tendency to gravitate toward internationally monitored and donor-funded programs. A similar experience is noted in many low-and-middle-income countries (LMICs), where NCDs are grossly underfunded and receive variable and often fleeting attention from government leaders [16, 21–23].

Such limited attention to NCDs at the PHC level may also be linked to inadequate technical capacity for NCDs among sub-national government bodies, little support from authorities at the national level, and inadequately delineated decision-making roles between national and sub-national actors. Although Ethiopia's decentralised health system is instrumental in improving frontline health care, our findings showed that for NCDs, there was a considerable mismatch between the policy responsibilities of subnational actors and the actual resources and other support mechanisms allocated to them [24]. Misalignment between responsibilities allocated to sub-national governments and the resources available to them is one of the challenges of decentralisation. This is not unique to Ethiopia; it mirrors evidence from, for example, Nigeria, China and Organisation for Economic Co-operation and Development (OECD) countries where the lack of sufficient appreciation for decentralisation has sometimes led to service fragmentation and health system performance variations [25–27]. Our findings indicate that successful integration of NCD programs in decentralised health systems requires continued technical support, efficient resource generation and distribution mechanisms and effective coordination across multiple levels. It also requires creating an enabling environment with explicit guidelines on engagement rules and a clear delineation of responsibilities between the various players, and an overall financial commitment to support the system.

As experience from HIV and MCH in Ethiopia and elsewhere showed, one enabling factor is the creation of a dedicated unit and/or focal person for NCDs at different health system levels [13, 23, 28]. Creating such a structure for NCDs could help strengthen governance, delineate responsibilities, and prioritise NCDs within the health system. In the Ethiopian context, replicating the national Joint Steering Committee at the regional and woreda levels and creating multi-sectoral coordination structure could prioritise the NCD agenda and enhance program implementation across the health system [13, 29]. Ideally, senior government officials should chair the multi-sectoral coordination committee, with MOH assuming a coordination role and given the power to drive NCD efforts in the country [13].

Similarly, as seen from the successful experience in other programs such as HIV, community engagement, particularly civil society, the private sector, community, and religious leaders, is critical in increasing demand generation and uptake of NCD interventions [16].

Existing health facility boards should be strengthened and empowered to foster community engagement at the local level [14]. A narrative review that included papers from 25 countries indicates that in decentralised health systems, local health boards or committees play a major role in improving service acceptance and satisfaction among service users; they also help to enhance community perception and accountability at lower levels of government [30]. Such structures and mechanisms are also necessary to promote the NCD agenda and its awareness locally, and to facilitate resource mobilisation, greater efficiency and accountability at all levels.

Our study participants identified funding as a major bottleneck for access to and provision of NCD services. As in other countries, donor funding for health in Ethiopia prioritises HIV, tuberculosis, and maternal health programs [21, 22, 31]. Domestic resources for NCDs are scant, resulting in costs borne primarily by patients and exacerbating the risk of catastrophic household debt [16, 32, 33]. The investment case for NCDs conducted by the Ethiopian government is a step in the right direction to identify funding gaps and advocate for more resources [34]. However, a concerted effort is required to build a sustainable health financing scheme in the country, primarily through increasing financial protection, reducing waste and achieving greater efficiency in service provision. The country's community health insurance system needs strengthening. This includes increasing population coverage, providing greater levels of co-payment relief to patients and expanding the types of NCD services covered [34, 35]. In parallel, costs to the health system could be lessened by broadening prevention strategies that help reduce service demand, leverage existing programs and achieve economies of scale within the health system [13, 16].

Another key strategy to containing costs is reducing over-reliance on hospital-based care. Again, there are lessons to be drawn from the successful experiences of HIV, tuberculosis, and MCH programs registered in several LMICs [36, 37]. NCD programs need to be expanded to all primary health care facilities and fully integrated and supported by appropriate referral and regular follow-up [37]. Such a strategy contains costs and addresses the mainly fragmented and vertically organised nature of NCD programs. Several countries have taken steps to integrate NCD services into HIV care delivery that increased coverage of NCD services and potential improvements in health outcomes for those with NCDs [38]. In Ethiopia, the success of HIV, tuberculosis, and MCH programs was linked to HEP's integrated community-based model of care [36, 37, 39, 40]. This involved task-sharing, simplifying protocols and guidelines, adopting standardised essential drugs list and diagnostic packages, and harmonising recording and reporting systems. It also included building a comprehensive model of care incorporating prevention, screening, diagnosis, and management of common diseases, along with a strong follow-up and referral system to cater for higher needs. While a similar model of care has been proposed for NCDs in Ethiopia and other LMICs, progress to provide NCD services has been slow, and this is consistent with our study findings [4, 7, 22].

Evidence from many LMICs, including Ethiopia, suggests that task-sharing from physician to non-physician health workers (mostly nurse and community health workers) can address critical workforce shortages and increase access to essential NCD health services [41, 42]. In other settings, such workers have been crucial in delivering health education and conducting early detection, case management, and lifestyle interventions to manage NCDs in the community [43–45]. Despite this evidence from elsewhere and the expectation of Ethiopia's HEWs to deliver NCD services, we found that their role was limited [7]. One priority area we identified was strengthening the on-the-job training for HEWs and incorporating in the training modules NCD content to the same level as MCH, HIV and malaria contents. There also needs to be more active engagement with supervisors and managers in upskilling programs that integrate NCDs into routine training and supportive supervision.

The availability of affordable medicines and diagnostic tests is a complex challenge in sub-Saharan Africa [28]. Improving availability and access requires generating sufficient funds and lowering out-of-pocket costs [28]. The political commitment witnessed for cancer control by the government, which resulted in the significant expansion of cancer infrastructure, provides a precedent [16]. Alongside increased funding for essential medicines, the essential drug guideline at the PHC level needs revision. There is also a need for the NCD drug procurement processes to be devolved and made more efficient at lower-level facilities. The latter is particularly useful to address bottlenecks related to bureaucracy around procurement and distribution of medicines and technologies to PHC facilities.

Ethiopia has a functional HMIS and District Health Information System -II (DHIS-II); however, the NCD indicators captured in these systems are limited [22]. As in other African countries, Ethiopia lacks adequate systems for generating reliable morbidity and mortality data, including data on NCDs [28]. Recent initiatives in conducting population-based NCD STEPwise approach to surveillance (STEPS) survey need to be strengthened [16, 28] and complemented with reforming the routine health information systems to meet the growing needs for timely, complete and accurate data for decision making. For example, the experience of m-health technologies for MCH programs could be emulated for NCDs [46]. Moreover, while there is encouraging experience at the national level concerning the adoption of guidelines and protocols on NCDs and WHO's Package of Essential Non-communicable interventions (WHO PEN), such actions require to be replicated at the PHC level [16, 28]. Likewise, the Ethiopian Primary Health Care Clinical Guideline, adopted from the Practical Approach to Care Kit(PACK) should be expanded to include a full range of NCDs [47, 48].

This study is not without limitation. The current study was conducted among participants recruited from the national ministry of health and two regions purposefully selected for the research. The findings from these regions may not be generalised to other regional or sub-national entities due to the varying socio-economic and health system performance profiles. Second, the study data were collected in 2019, and much has changed since then in Ethiopia and the world. The COVID-19 pandemic and the current conflict in Ethiopia is likely to have re-oriented government priorities and disrupted health services previously available to the population. The study findings need to be interpreted, taking these facts in mind.

## Conclusion

Success in the prevention and control of NCDs through high quality PHC in Ethiopia requires building on past success with PHC reforms in other areas of health, while recognising challenges specific to NCDs. Key strategic priorities include: (1) cultivating strong political commitment and leadership with a focus on resource mobilisation and support for sub-national actors; (2) delineating responsibilities and creating clear governance arrangements such as instituting NCD units and assigning focal persons at subnational levels; (3) establishing multi-sectoral committees at national and sub-national levels to ensuring greater coordination between and within sectors; (4) strengthening community engagement mechanisms and empowering PHC facility boards; and (5) improving financial protection through an expanded community health insurance system.

Key operational priorities include: (1) greater investment in frontline PHC care and a shift away from hospital based services; (2) capacity building and supportive supervision for the PHC workforce; and (3) simplification of protocols and guidelines, supported by standardised essential drugs and diagnostic packages, harmonised recording and reporting systems, and facilitated referral mechanisms. Successful PHC program implementation requires optimising both strategic and operational priorities to achieve integrated and people-centred NCD

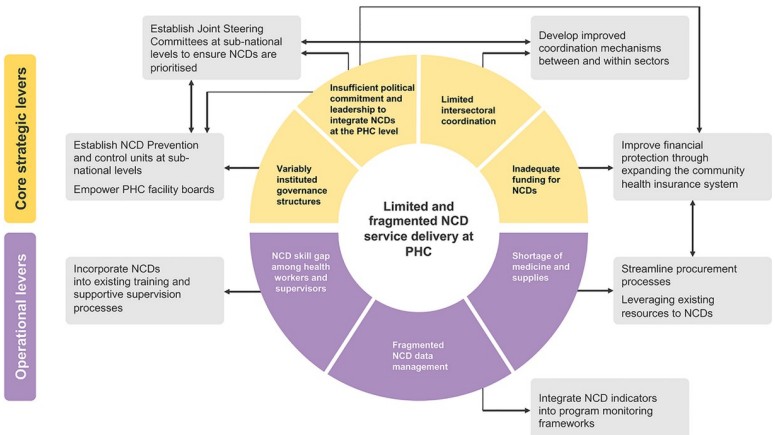

**Fig 2. Study findings to provide integrated and people-centred NCD prevention and control through PHC in Ethiopia, 2021.**

services in Ethiopia. A summary of the key study findings and proposed actions to provide integrated and people-centred NCD prevention and control through PHC are illustrated in Fig 2.

## Supporting information

**S1 File. English version interview guides.**
(DOCX)

## Acknowledgments

The authors would like to thank all the respondents who participated in the interviews. We also extend our acknowledgement to the research assistants who dedicated their time and expertise to this project.

## Author Contributions

**Conceptualization:** Azeb Gebresilassie Tesema, Seye Abimbola, Padmanesan Narasimhan, Rohina Joshi, David Peiris.

**Data curation:** Azeb Gebresilassie Tesema, Afework Mulugeta.

**Formal analysis:** Azeb Gebresilassie Tesema, Whenayon S. Ajisegiri, Rohina Joshi, David Peiris.

**Funding acquisition:** Rohina Joshi.

**Investigation:** Azeb Gebresilassie Tesema, Afework Mulugeta.

**Methodology:** Azeb Gebresilassie Tesema.

**Project administration:** Azeb Gebresilassie Tesema, Rohina Joshi, David Peiris.

**Software:** Azeb Gebresilassie Tesema.

**Supervision:** Seye Abimbola, Rohina Joshi, David Peiris.

**Validation:** Seye Abimbola, Rohina Joshi, David Peiris.

**Visualization:** Azeb Gebresilassie Tesema.

**Writing – original draft:** Azeb Gebresilassie Tesema.

**Writing – review & editing:** Azeb Gebresilassie Tesema, Seye Abimbola, Afework Mulugeta, Whenayon S. Ajisegiri, Padmanesan Narasimhan, Rohina Joshi, David Peiris.

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
