## [Decision Letter · Decision Letter 0]

2 Aug 2021

 PGPH-D-21-00231 HEALTH SYSTEM CAPACITY AND READINESS FOR DELIVERY OF INTEGRATED NON-COMMUNICABLE DISEASE SERVICES IN PRIMARY HEALTH CARE: A QUALITATIVE ANALYSIS OF THE ETHIOPIAN EXPERIENCE PLOS Global Public Health

Dear Dr. Tesema,

Thank you for submitting your manuscript to PLOS Global Public Health. After careful consideration, we feel that it has merit but does not fully meet PLOS Global Public Health’s publication criteria as it currently stands. Therefore, we invite you to submit a revised version of the manuscript that addresses the points raised during the review process.

We look forward to receiving your revised manuscript.

Kind regards,

Muhammad Fawad Rasool

Academic Editor

Journal Requirements:

Additional Editor Comments (if provided):

Reviewers' comments:

Reviewer's Responses to Questions

**Comments to the Author**

1. Does this manuscript meet PLOS Global Public Health’s publication criteria? Is the manuscript technically sound, and do the data support the conclusions? The manuscript must describe methodologically and ethically rigorous research with conclusions that are appropriately drawn based on the data presented.

Reviewer #1: Yes

Reviewer #2: Yes

2. Has the statistical analysis been performed appropriately and rigorously?

Reviewer #1: Yes

Reviewer #2: Yes

3. Have the authors made all data underlying the findings in their manuscript fully available (please refer to the Data Availability Statement at the start of the manuscript PDF file)?

Reviewer #1: Yes

Reviewer #2: Yes

4. Is the manuscript presented in an intelligible fashion and written in standard English?

Reviewer #1: Yes

Reviewer #2: Yes

5. Review Comments to the Author

Reviewer #1: There are only two notes for minor revision:

A- In the abstract method, the sentence “in two regions and Addis Ababa, Ethiopia” after two regions, the “and” does not make sense.

B- In the main method body, study settings section, please clarify “woredas” as it is not an English word.

1. The study presents the results of the original research.

Yes, as it was shown throughout the article, the originality of the research. Plus, it has not been published elsewhere.

2. Results reported have not been published elsewhere.

Yes, it was selective and a new publication.

3. Experiments, statistics, and other analyses are performed to a high technical standard and are described in sufficient detail.

The analysis was qualitative with 22 participants. The 22 participants required details were present. Also, the details of the experiments are informative.

4. Conclusions are presented in an appropriate fashion and are supported by the data.

The conclusion is informative and thorough.

5. The article is presented in an intelligible fashion and is written in standard English.

Please fix the following:

1- In the abstract method, the sentence “in two regions and Addis Ababa, Ethiopia” after two regions, the “and” does not make sense.

2- In the method main body, study settings section; please clarify “woredas” as it is not an English word

6. The research meets all applicable standards for the ethics of experimentation and research integrity.

Yes, the authors not only provide approval from their local area where the study was conducted but also from Australia.

The committee board and approval number: National

Research and Research Ethics

office under the

Ethiopian Public Health

Institute (EPHI-IRB-194-2019) The committee board and approval number:: The

University of New

South Wales (UNSW)

Human Research Ethics Committee

(HC190014)

7. The article adheres to appropriate reporting guidelines and community standards for data availability.

Yes, the authors had followed the available guidelines and made the data available upon request.

Reviewer #2: To me although there is some needs to know more about the Ethiopian heath numerical indicators to understand the essentiality of the study and the burden of NCD in the country but the research well designed and has value to study. also I expect to get the answers for three main objectives more typically in the conclusion.

6. PLOS authors have the option to publish the peer review history of their article (what does this mean?). If published, this will include your full peer review and any attached files.

**Do you want your identity to be public for this peer review?** For information about this choice, including consent withdrawal, please see our Privacy Policy.

Reviewer #1: **Yes: **Aliah Aldahash

Reviewer #2: No

---

## [Editor Report · Decision Letter 1]

15 Sep 2021

HEALTH SYSTEM CAPACITY AND READINESS FOR DELIVERY OF INTEGRATED NON-COMMUNICABLE DISEASE SERVICES IN PRIMARY HEALTH CARE: A QUALITATIVE ANALYSIS OF THE ETHIOPIAN EXPERIENCE

PGPH-D-21-00231R1

Dear Dr. Tesema,

We're pleased to inform you that your manuscript has been judged scientifically suitable for publication and will be formally accepted for publication once it meets all outstanding technical requirements.

Within one week, you'll receive an e-mail detailing the required amendments. When these have been addressed, you'll receive a formal acceptance letter and your manuscript will be scheduled for publication.

An invoice for payment will follow shortly after the formal acceptance. To ensure an efficient process, please log into Editorial Manager at https://www.editorialmanager.com/pgph/ click the 'Update My Information' link at the top of the page, and double check that your user information is up-to-date. If you have any billing related questions, please contact our Author Billing department directly at authorbilling@plos.org.

Kind regards,

Muhammad Fawad Rasool

Academic Editor